# An ERF Transcription Factor Gene from *Malus baccata* (L.) Borkh, *MbERF11*, Affects Cold and Salt Stress Tolerance in *Arabidopsis*

**Deguo Han, Jiaxin Han, Guohui Yang, Shuang Wang, Tianlong Xu and Wenhui Li \***

Key Laboratory of Biology and Genetic Improvement of Horticultural Crops (Northeast Region), Ministry of Agriculture and Rural Affairs, College of Horticulture & Landscape Architecture, Northeast Agricultural University, Harbin 150030, China; deguohan_neau@126.com (D.H.); A02140301@163.com (J.H.); yangguohui_neau@126.com (G.Y.); ws18045312966@163.com (S.W.); 18846829610@163.com (T.X.)

**\*** Correspondence: lwh_neau@126.com; Tel.: +86-4515-5190-781

**Abstract:** Apple, as one of the most important economic forest tree species, is widely grown in the world. Abiotic stress, such as low temperature and high salt, affect apple growth and development. Ethylene response factors (ERFs) are widely involved in the responses of plants to biotic and abiotic stresses. In this study, a new ethylene response factor gene was isolated from *Malus baccata* (L.) Borkh and designated as *MbERF11*. The *MbERF11* gene encoded a protein of 160 amino acid residues with a theoretical isoelectric point of 9.27 and a predicated molecular mass of 17.97 kDa. Subcellular localization showed that MbERF11 was localized to the nucleus. The expression of *MbERF11* was enriched in root and stem, and was highly affected by cold, salt, and ethylene treatments in *M. baccata* seedlings. When *MbERF11* was introduced into *Arabidopsis thaliana*, it greatly increased the cold and salt tolerance in transgenic plant. Increased expression of *MbERF11* in transgenic *A. thaliana* also resulted in higher activities of superoxide dismutase (SOD), peroxidase (POD), and catalase (CAT), higher contents of proline and chlorophyll, while malondialdehyde (MDA) content was lower, especially in response to cold and salt stress. Therefore, these results suggest that *MbERF11* probably plays an important role in the response to cold and salt stress in *Arabidopsis* by enhancing the scavenging capability for reactive oxygen species (ROS).

**Keywords:** *Malus baccata*; *MbERF11*; cold stress; salt stress; transgenic plant

## 1. Introduction

During growth and development, plants are frequently exposed to various abiotic stresses, such as drought, cold, salt, heat, and nutrient deprivation. Under different environmental stresses, plants have developed different adaptable mechanisms to ensure their normal growth and development [1,2]. The response mechanism of plants to abiotic stresses was regulated by multiple signaling pathways [3]. In these processes, transcription factors (TFs) play a key role in the interaction of these signaling pathways [4–7]. TFs, also known as trans-acting factors, are a class of DNA-binding proteins that specifically bind to cis-acting elements. Interactions between transcription factors and cis-acting elements or other proteins can be transcriptional activation or inhibition. Studies have clarified that TF genes are abundantly present in plants, and can regulate plant growth and metabolism [8,9].

The APETALA2/ethylene responsive factor (AP2/ERF) TFs are widely distributed in different types of plants [10]. About 145 AP2/ERF TF genes have been isolated from *A. thaliana* [11], and 167 AP2/ERF TFs were also found in Oryza sativa [12]. The AP2/ERF TF family may have evolved from HNHAP2 endonucleases in bacteria and viruses [13]. The AP2/ERF genes were widely isolated and studied from different plants, such as *APETALA2*, *AINTEGUMENTA* and *AtCBF1* from *Arabidopsis* [14–16],

*ThCRF1* from *Tamarix hispida* [17], and *CsERF025* from cucumber [18]. The AP2/ERF TFs were found to participate in almost every process of plant growth and development, especially in response to biotic or abiotic stresses in plants [19].

As a class of TFs, AP2/ERF TFs usually contain an AP2/ERF domain consisting of about 60 amino acid residues [20]. This domain was firstly discovered from APETALA2 homologues of *A. thaliana* [21], a similar sequence was also found in tobacco [22]. According to the different numbers of AP2/ERF domains contained in AP2/ERF, the AP2/ERF TFs were divided into two subfamilies. Among them, one subfamily is called ERF containing one AP2/ERF domain. The other subfamily is called the AP2 subfamily containing two AP2/ERF domains. The ERF subfamily can be further divided into DREB (dehydration responsive element binding protein), ERF and other three categories [23]. In the AP2/ERF domain of DREB subfamily, the 14th and 19th amino acids are valine (V) and glutamic acid (E), respectively, while they are alanine and aspartate in the AP2/ERF domain of ERF subfamily [24].

ERF TFs are widely involved in biotic and abiotic stress responses, which play important roles in drought, high salt and low temperature tolerance, as well as plant development, hormone response and other regulatory networks [24,25]. Previous researches found ERF TFs were also involved in organ development, cell division, differentiation, flower development and fruit maturation in plants [26,27]. Overexpression of *Sl-ERF2* gene in tomato could activate *Sl-Man2*, a mannanase-encoding gene, and result in the premature germination of tomato immature seeds [28]. The *Sub1A/C* gene in *O. sativa* participates in plant growth and metabolism [29]. The *MdERF1/2* genes in *Malus domestica* are associated with fruit ripening [30]. However, these studies mostly focused on model plants or crops, and the roles of the ERF TFs genes in *Malus* plant stress responses were less well known.

*M. baccata* is widely used as an apple rootstock in northern China, and also as a source of forestry wood or greening tree species. Due to its high resistance to low temperature and drought, M. baccata is also used as a breeding material for cold and drought resistance [31]. From the transcriptome analysis of *M. baccata* seedlings under cold and/or drought stresses (results not presented here), we found the *MbERF11* level was significantly up-regulated under both stresses. More importantly, through NCBI blast (https://blast.ncbi.nlm.nih.gov/Blast.cgi) of *MbERF11* gene, we found that the closest *Arabidopsis ERF* gene is *AtERF7*, which is a famous ERF TF gene involved in drought stress through ABA signal transduction [32]. To better understand the role of ERF TFs genes involved in low temperature and salt stress, and to provide potentially genetic resources for the improvement of the drought resistance of *Malus* plant, a new ERF TFs gene was isolated from *M. baccata* and designated as *MbERF11*. Moreover, it was found that the tolerance of transgenic *A. thaliana* to cold and salt stress was increased because of the overexpression of *MbERF11*.

## 2. Materials and Methods

### 2.1. Plant Material and Growth Conditions

The *M. baccata* test-tube seedlings were rapidly propagated in MS growth medium (containing 0.6 mg/L IBA + 0.6 mg/L 6-BA) for 30 days. Then they were transplanted to MS + 1.2 mg/L IBA for 45 days for rooting. Finally, the seedlings were transferred to Hoagland solution for 40 days for growth. The solution was changed three times per week. The temperature of the culture chamber was maintained at 20 °C and the relative humidity was maintained at about 85%. When the test-tube seedlings grew to 8–9 leaves (completely expanded), a part of them was placed in Hoagland solution with a NaCl concentration of 200 μM and pH 5.8 for salt stress treatment. The other part of them was placed in Hoagland solution at 4 °C for cold stress treatment. Test-tube seedlings incubated in Hoagland solution at 20 °C were used as control. The unexpanded young leaf, the completely unfolded mature leaf, the phloem at the second and third node stem segments, and the newly emerged root were taken as samples. The samples of all control and treated plants were sealed after treatments of respectively 0, 2, 4, 8, 12, 24, and 48 h, immediately frozen in liquid nitrogen, and then stored at −80 °C for RNA extraction.

## 2.2. The qRT-PCR Expression Analysis of MbERF11

Total RNA was respectively extracted from young leaf, mature leaf, new root, and stem using the EasyPure Plant RNA Kit (TransGen, Beijing). Synthesis of cDNA first strands with TransGen's Trans Script® First-Strand cDNA Synthesis Super Mix (TransGen, Beijing). The whole sequence of *MbERF11* was obtained by PCR, with the first strand cDNA of *M. baccata* as a template. A pair of primers (*MbERF11*-F and *MbERF11*-R, Table 1) were designed based on the homologous regions of *MdERF011* (MDP0000258562) to amplify the full-length cDNA sequence. The obtained DNA fragments were gel purified and cloned into the pEasy-T1 vector (TransGen) and sequenced (BGI, Beijing).

**Table 1.** Primers used in this study.

| Name of Primer | Sequence of Primer (5′→3′) | Purpose |
|---|---|---|
| *MbERF11*-F | ATGGAAGGAGATTACTGCTGCT | Cloning |
| *MbERF11*-R | TTAACTTTCATCGGAGTTTTCTGGG | Cloning |
| *ACTIN*-F | ACACGGGGAGGTAGTGACAA | qPCR |
| *ACTIN*-R | CCTCCAATGGATCCTCGTTA | qPCR |
| *GAP*-F | GTCGTACTACTGGTATCGTT | qPCR |
| *GAP*-R | TCATAGTCAAGAGCAATGTA | qPCR |
| MF | TGGAGAAGCGTAAGCATCCC | qPCR |
| MR | CGTCGTCTTGGAATACAAGCT | qPCR |
| *Site*-F | GAGCTCATGGAAGGAGATTACTGCT | Add *Sac*I site |
| *Site*-R | CCTAGGACTTTCATCGGAGTTTTCTG | Add *Bam*HI site |

The qRT-PCR expression analysis of *MbERF11* was performed according to method of Han et al. [33]. The *Malus ACTIN* gene (AB638619.1) amplified from *M. baccata* tissues was as control, which was stably expressed under various conditions [34]. We designed the primers (*ACTIN*-F and *ACTIN*-R, Table 1) from the sequences and published in the GenBank databases. The primers (MF and MR, Table 1) of *MbERF11* were designed from partial sequences cloned in this study for qRT-PCR detection. The thermal cycling program was one initial cycle of 94 °C for 30 s, followed by 40 cycles of 94 °C for 15 s, and 55 °C for 30 s. The relative transcription level data was analyzed by the Pfaffl method [35].

## 2.3. Subcellular Localization Analysis of MbERF11 Protein

The *MbERF11* ORF was cloned into the *Sac*I and *Bam*HI sites of the pSAT6-GFP-N1 vector. This vector contains a modified red-shifted green fluorescent protein (GFP) at *Sac*I–*Bam*HI sites. The *MbERF11*-GFP construct was transformed into onion (*Allium cepa*) epidermal cells by particle bombardment [36]. The DAPI staining was used as a nucleus marker for nucleus detection. The transient expression of the MbERF11–GFP fusion protein was observed by confocal microscopy.

## 2.4. A. Thaliana Transformation

To construct an expression vector for the transformation of *A. thaliana*, restriction enzyme cut sites of *Sac*I and *Bam*HI at 5′ and 3′ ends of the *MbERF11* cDNA was respectively added by PCR with the primers (*Site*-F, *Site*-R, Table 1). To construct the PCAM3301-*MbERF11* vector, PCAM3301 and the products of PCR were digested by *Sac*I and *Bam*HI, and linked together by T4 DNA ligase. The *MbERF11* gene driven by the CaMV 35S promoter and the vecror (only PCAM3301) were introduced into *A. thaliana* by Agrobacterium-mediated LBA4404 transformation [37]. Columbia ecotype *A. thaliana* plants were transformed using the vacuum infiltration method. Transformants (transgenic lines and vector line) were selected on MS medium containing 6 mg/mL glufosinate. The transgenic lines (roots used as materials) were confirmed by qRT-PCR analysis with wild type (WT) and vacant-vector line (VL) as control. T3 generation plants were used for further analysis.

### 2.5. Determination Survival Rates Under Cold and/or Salt Stress Treatment in Transgenic Arabidopsis

Wild-type *A. thaliana* (WT), vacant-vector line (VL, the line only transformed with vacant vector) and three *MbERF11* transgenic lines (S2, S6, S7) were respectively sown in culture medium, and transferred to nutrient soil for two weeks after 10 days. The WT, VL and T3 transgenic *A. thaliana* were cultured under control condition, low temperature treatment (−4 °C) for 12 h, and high salinity stress (200 mM NaCl) for 7 d, respectively, after which their survival rates were recorded with 15 nutrition pots.

### 2.6. Detection of the Contents of Chlorophyll, MDA and Proline and the Activity of SOD, POD and CAT

All the materials of different lines above were collected for measurements. The chlorophyll content was determined with method of Li et al. [38]. The proline content was measured according to the spectrophotometric method [39]. The MDA content and the activities of SOD, POD, and CAT were measured according to the protocol described by Shin et al. [40].

### 2.7. Statistical Analysis

DPS 7.05 data processing system software was used for one-way analysis of variance (ANOVA). All experiments were repeated for three times and the standard errors (±SE) were measured, respectively. Statistical differences were referred to as significant * $p \le 0.05$, ** $p \le 0.01$.

## 3. Results

### 3.1. Isolation of MbERF11 Gene from M. Baccata

The ProtParam analysis (http://www.expasy.org/tools/protparam.html) showed that the *MbERF11* gene encodes 160 amino acids (Figure 1). The theoretical molecular mass of MbERF11 is 17.97 kDa, with theoretical isoelectric point 9.27 and the average hydrophilicity coefficient −0.995. The underlined part of Figure 1 is the conserved sequence of the AP2/ERF family, which contains two conserved elements, namely YRG and RAYD. The 14th and 19th amino acids of the conserved sequence are valine and glutamic respectively, indicating that it belongs to the DREB subfamily in the AP2/ERF family.

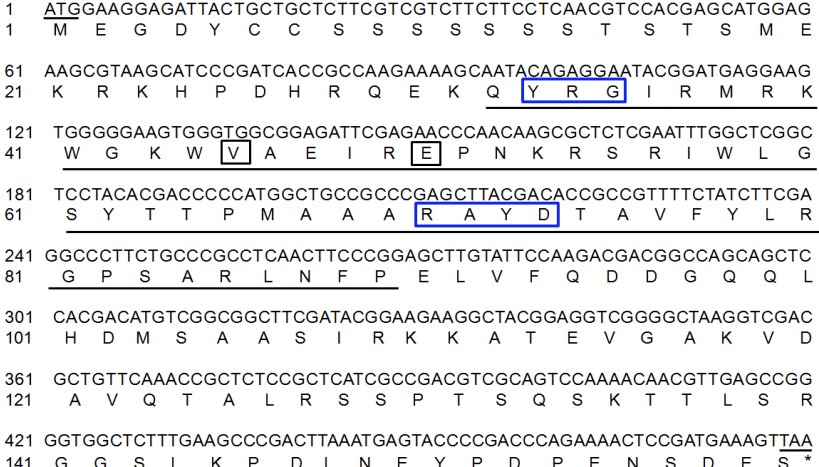

**Figure 1.** Nucleotide and deduced amino acid sequences of *MbERF11* gene. Underlines indicate conserved amino acid sequences. Black boxes indicate specific amino acids of the AP2/ERF domain. Blue boxes indicate conserved elements.

### 3.2. Phylogenetic Relationship of MbERF11 with Other ERF Proteins

To explore the evolutionary relationship among plant ethylene response factors, DNAman was used to compare MbERF11 with other 13 ERF proteins of different species. The phylogenetic tree showed that MbERF11 contains an AP2/ERF conserved domain consisting of 58 amino acid residues. This conserved sequence is the characteristic sequence of the ERF TF in plants (Figure 2A). As shown in Figure 2A, the TF of ERF family has higher homology in the conserved domain. There are also many changes in the non-conserved domain, which is consistent with the characteristics of the TF.

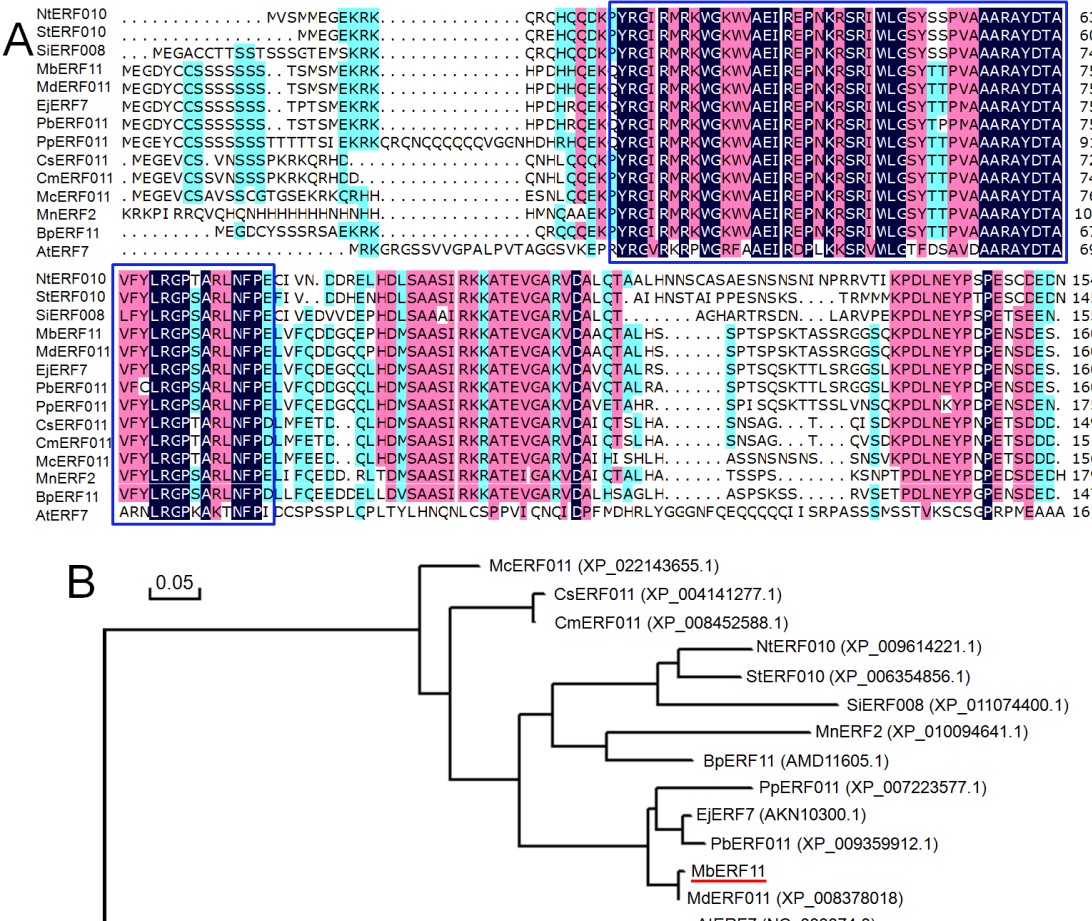

**Figure 2.** Comparison and phylogenetic relationship of MbERF11 with ethylene-responsive transcription factor proteins from other species. (**A**) Comparison completes alignment of MbERF11 with other plant ethylene-responsive transcription factor proteins. Conserved domains are shown in blue boxes. Positions containing identical residues are shaded in navy blue, while conservative residues are shown in green. (**B**) Phylogenetic tree analysis of MbERF11 and other plant ethylene-responsive transcription factor proteins. The tree was constructed by the neighbour-joining method with MEGA 7 (http://www.megasoftware.net). The gene accession numbers are listed in Figure 2B.

The homologous phylogenetic tree showed that MbERF11, MdERF011 (XP_008378018, *Malus domestica*), EjERF7 (AKN10300.1, *Eriobotrya japonica*), PbERF011 (XP_009359912.1, *Pyrus bretschneideri*) and PpERF011 (XP_007223577.1, *Prunus persica*) clustered together. NtERF010 (XP_009614221.1, *Nicotiana tomentosiformis*), StERF010 (XP_006354856.1, *Solanum tuberosum*), SiERF008 (XP_011074400.1, *Sesamum indicum*), MnERF2 (XP_010094641.1, *Morus notabilis*), and BpERF11 (AMD11605.1, from *Betula platyphylla*) were grouped into the second cluster, followed by CsERF011 (XP_004141277.1, *Cucumis sativus*), CmERF011 (XP_008452588.1, *Cucumis melo*), and McERF011 (XP_022143655.1, *Momordica charantia*). However, AtERF7 (NC_003074.8) was grouped into another cluster on its own (Figure 2B).

### 3.3. MbERF11 was Localized to the Nucleus

As shown in Figure 3, the MbERF11–GFP fusion protein is targeted to nucleus (Figure 3E) with 4′, 6-diamidino-2-phenylindole (DAPI) staining (Figure 3F), whereas the control GFP alone is distributed throughout the cytoplasm (Figure 3B). These results determined that MbERF11 is a nucleus localized protein.

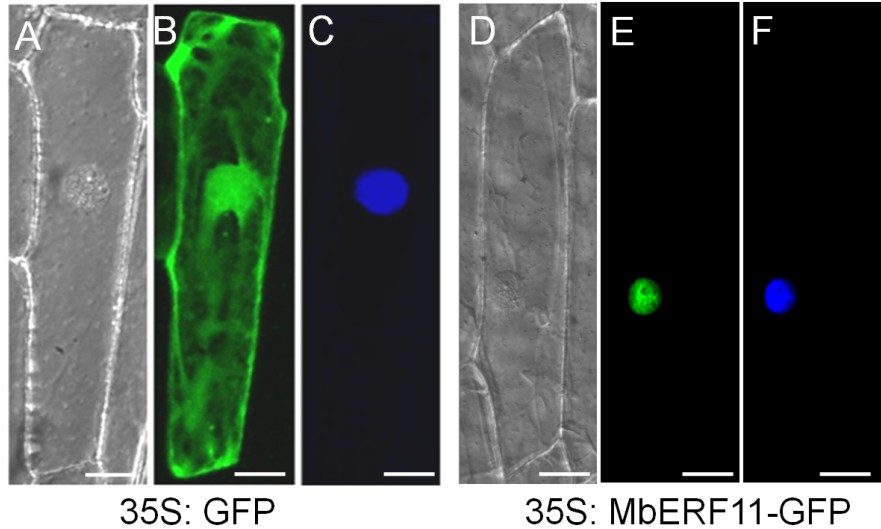

**Figure 3.** Subcellular localization of MbERF11 protein. The transient vector harboring 35S-GFP and 35S-*MbERF11*-GFP cassettes were transformed into onion epidermal cells by particle bombardment. Transient expressions of green fluorescent protein 35S-GFP (**B**) and 35S-*MbERF11*-GFP (**E**) translational product were visualized in onion epidermal cells by fluorescence microscopy. Onion epidermal cells of 35S-GFP (**C**) and 35S-*MbERF11*-GFP (**F**) stained with DAPI for 24 h. (**A**,**D**) were taken in the bright light. Scale bar corresponds to 5 μm.

### 3.4. Expression Analysis of MbERF11 in M. Baccata

As shown in Figure 4A, in control condition, the expression level of *MbERF11* in *M. baccata* seedlings was higher in root and stem, while very low in leaf. When dealt with salt (200 mM NaCl), cold (4 °C), and ethephon treatments (500 μL/L, the ratio of ethylene:water is 1:2000), the expression level of *MbERF11* in young leaf of *M. baccata* increased quickly, reached maximum at 12 h, 24 h, and 4 h, respectively, and then decreased (Figure 4B). The expression level of *MbERF11* in root had a similar trend, which reached the maximum at 8 h, 12 h, and 2 h, respectively, then decreased slightly (Figure 4C).

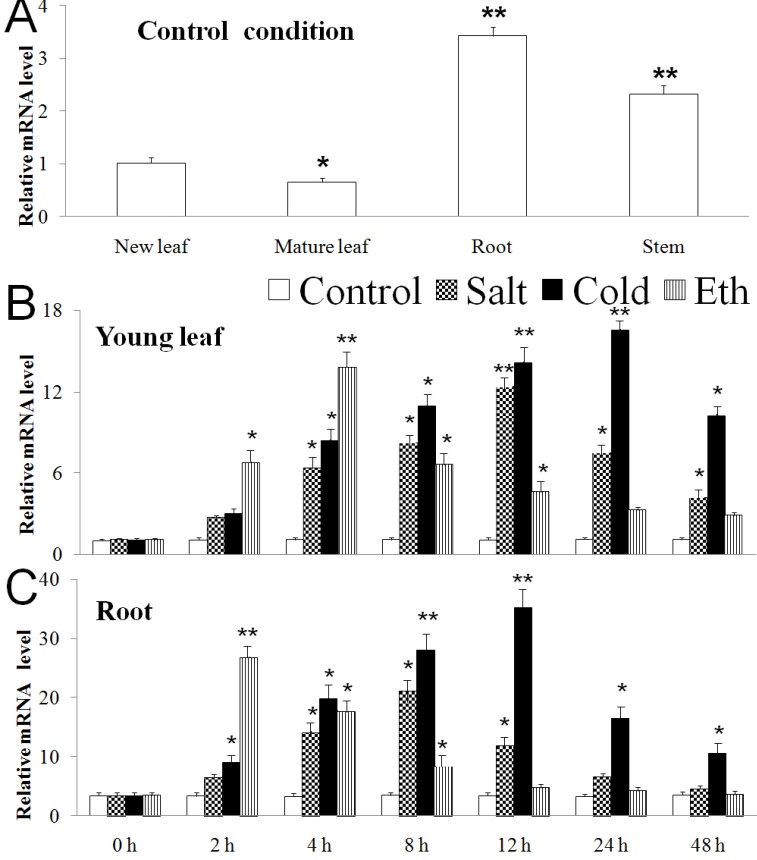

**Figure 4.** Time-course expression patterns of *MbERF11* in *Malus baccata* using qRT-PCR methord. (**A**) Expression patterns of *MbERF11* in young leaf (partly expanded), mature leaf (fully expanded), root and stem in normal condition (room temperature and normal nutrient solution). The expression level of young leaf was as control. (**B**,**C**) Expression patterns of *MbERF11* in control, salt (200 μM), cold (4 °C) and ethephon (500 μL/L) in young leaf (**B**), and root (**C**) at the following time points: 0 h, 2 h, 4 h, 8 h, 12 h, 24 h and 48 h. The reference genes *MdACTIN* and *MdGAPDH* were used as controls in this study. The error bars represent standard deviation. Asterisks above the error bars indicate a significant difference between the treatment and control (0 h) using Student's *t* test (* $p \leq 0.05$; ** $p \leq 0.01$).

### 3.5. Overexpression of MbERF11 in A. Thaliana Contributed to Low Temperature Stress Tolerance

To study the role of *MbERF11* in cold and salt stress responses, *MbERF11* gene was transformed into *A. thaliana* under the control of the CaMV 35S promoter. Among 12 transformed lines, seven of them (S1, S2, S4, S6, S7, S9, and S10) were confirmed by qRT-PCR analysis with wild type (WT) and vacant-vector line (VL) as control (Figure 5A).

As shown in Figure 5B, no significant difference in appearance was found among all the *A. thaliana* lines (WT, VL, S2, S6, and S7) under control condition (Cold 0h). However, when dealt with low temperature (−4 °C) stress for 12 h (Cold 12 h), the transgenic plants (S2, S6 and S7) look much healthier than WT and VL. There were no differences in appearance between WT and VL under control condition and cold stress.

Under control condition, there was no significant difference in the survival rates among all *A. thaliana* lines (WT, VL, S2, S6 and S7). However, when dealt with cold stress, the survival rates of WT and VL *A. thaliana* were only 16.7% and 18.3%, while the transgenic plants of S2, S6, and S7 were 78.7%, 75.1%, and 81.3%, respectively. The survival rates of transgenic plants were significantly higher than those of WT and VL lines under low temperature treatments (Figure 5C).

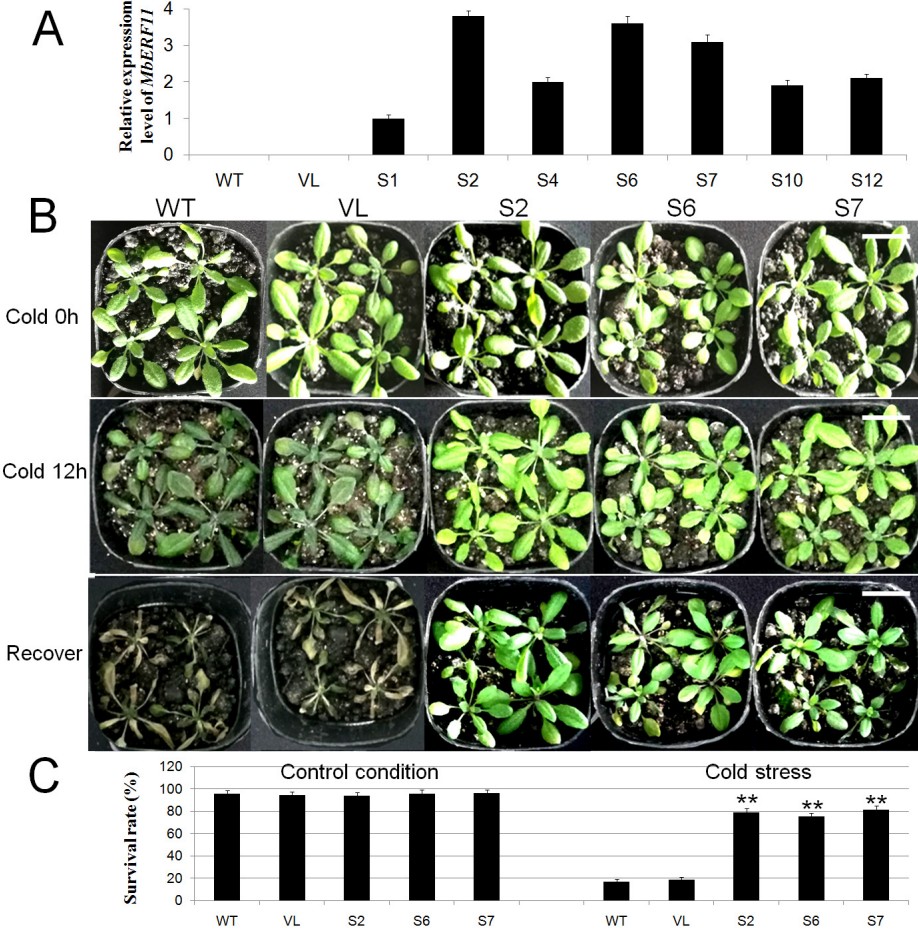

**Figure 5.** Overexpression of *MbERF11* in *Arabidopsis* improved cold tolerance. (**A**) Transgenic *Arabidopsis* qRT-PCR validation. (**B**) Phenotypic map of *Arabidopsis* under control condition (Cold 0 h), cold stress (Cold 12 h) and recover. All the test lines, including the $T_3$ transgenic *A. thaliana* (S2, S6, and S7), WT and VL were dealt with low temperature stress (−4 °C) for 12 h, and then with control temperature (20 °C) 1 h for recover. Scale bar corresponds to 3 cm. (**C**) The survival rates of transgenic and WT *Arabidopsis* under control condition and low temperature stress. Extremely significant differences between transgenic *Arabidopsis* (S2, S6, and S7), vacant-vector line (VL) and WT line were shown by the *t* test, ** $p \leq 0.01$.

As shown in Figure 6, for the contents of chlorophyll, MDA and proline, as well as the activities of SOD, POD, and CAT, these was no significant difference among all the *A. thaliana* lines, i.e., S2, S6, S7, WT and VL plants under control conditions (Cold 0 h). However, when dealt with cold treatment (−4 °C) for 12 h (Cold 12 h), the activities of SOD, POD, and CAT, the chlorophyll and proline contents were higher than those in WT and VL. However, the contents of MDA in transgenic *A. thaliana* (S2, S6, and S7) were significantly lower than those in WT and VL (Figure 6).

### 3.6. Overexpression of MbERF11 in A. Thaliana Contributed to Improved Salt Stress Tolerance

The WT, VL and transgenic lines (S2, S6, and S7) of *A. thaliana* were treated with 200 mM NaCl solution daily for seven days, and then the phenotype of each line was observed. As shown in Figure 7A, the transgenic lines (S2, S6, and S7), WT and VL plants all grew well under control condition (Salt 0 d). However, when dealt with salt stress for 7 days (Salt 7 d), the transgenic *A. thaliana* (S2, S6 and S7) had better appearance than WT and VL plants.

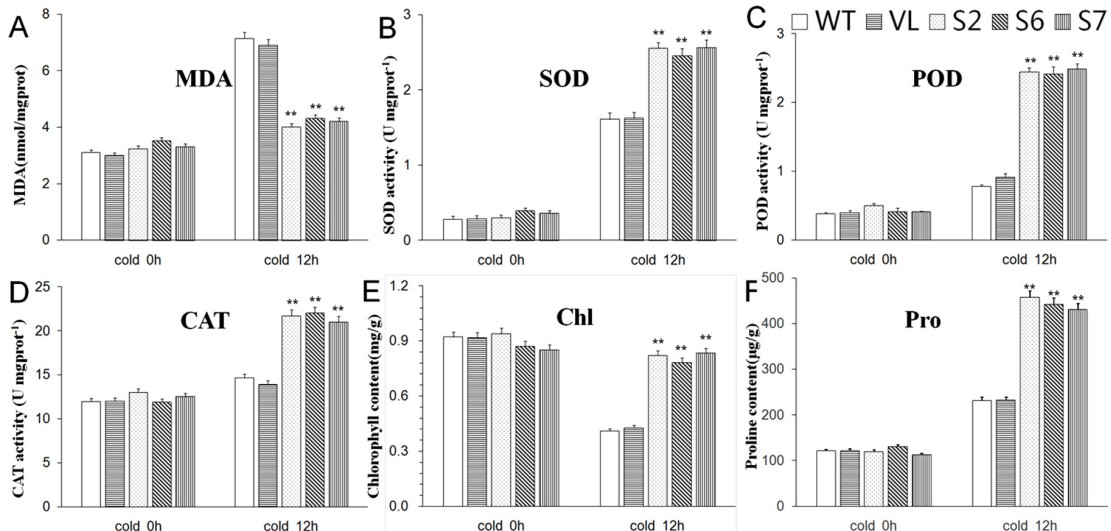

**Figure 6.** The levels of malondialdehyde content ((**A**) MDA); superoxide dismutase activity ((**B**), SOD); peroxidase activity ((**C**), POD); catalase activity ((**D**), CAT); chlorophyll content ((**E**), Chl) and proline content ((**F**), Pro) Proline in WT, VL (vacant-vector line) and *MbERF11*-OE (S2, S6, and S7) *Arabidopsis* under control condition (Cold 0 h) and cold stresses (Cold 12 h). Asterisks above the error bars indicate an extremely significant difference between the transgenic and WT plants with Student's *t* test (** $p \le 0.01$). The level of each index in WT was as control.

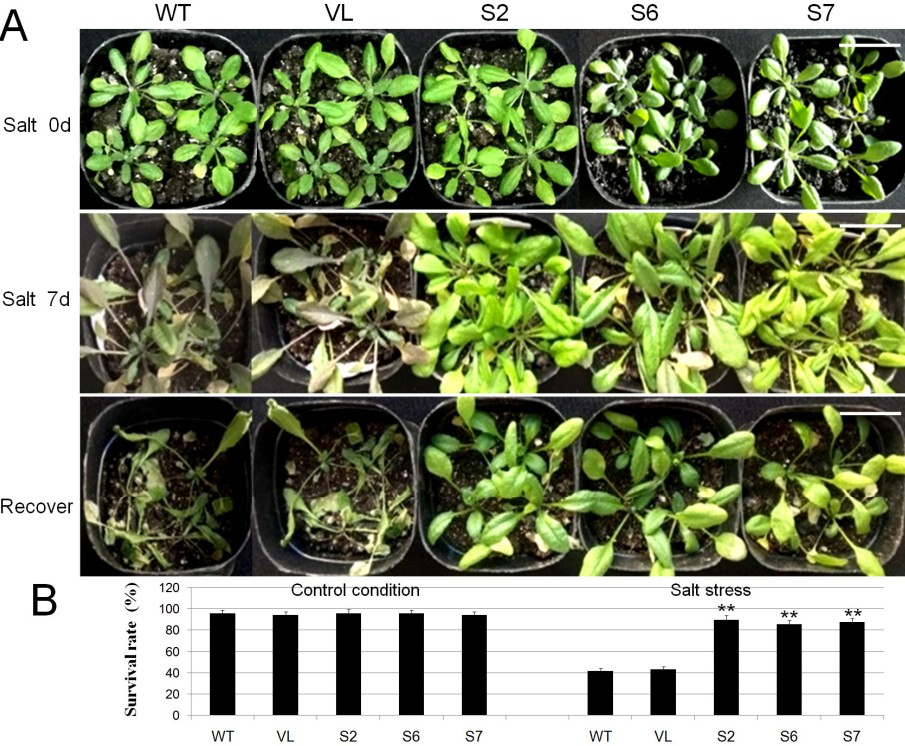

**Figure 7.** Overexpression of *MbERF11* in *Arabidopsis* improved salt tolerance in transgenic plants. (**A**) Phenotypic map of *Arabidopsis* under control condition (Salt 0 d), salt stress (Salt 7 d) and recover. All the test lines, including the $T_3$ transgenic *A. thaliana* (S2, S6, and S7), WT and VL were dealt with salt stress (200 mM NaCl) for 7 d, and then with control water management 3 d for recover. Scale bar corresponds to 3 cm. (**B**) The survival rate of *Arabidopsis* under control condition and salt stress. Extremely significant differences between transgenic *Arabidopsis* (S2, S6, and S7), VL and WT line were shown by the *t* test, ** $p \le 0.01$.

Under control condition, there was no significant difference in the survival rates of all *A. thaliana* lines (WT, VL, S2, S6, and S7). However, when dealt with salt stress for 7 days, the survival rates of WT and VL plants were only 41.9% and 43.2%, while the transgenic lines of S2, S6, and S7 were 89.7%, 85.8% and 87.6%, respectively. The survival rates of the transgenic plants under salt stress were significantly higher than those of WT and VL plants (Figure 7B).

As shown in Figure 8, when treated with salt stress (Salt 7 d), overexpression of *MbERF11* in transgenic *A. thaliana* resulted in lower MDA contents, higher levels of chlorophyll and proline contents, as well as higher activities of SOD, POD, and CAT than those of WT and VL plants. However, for the indices above, these was no significant difference among the entire test lines (WT, VL, S2, S6, and S7) under control condition (Salt 0 d).

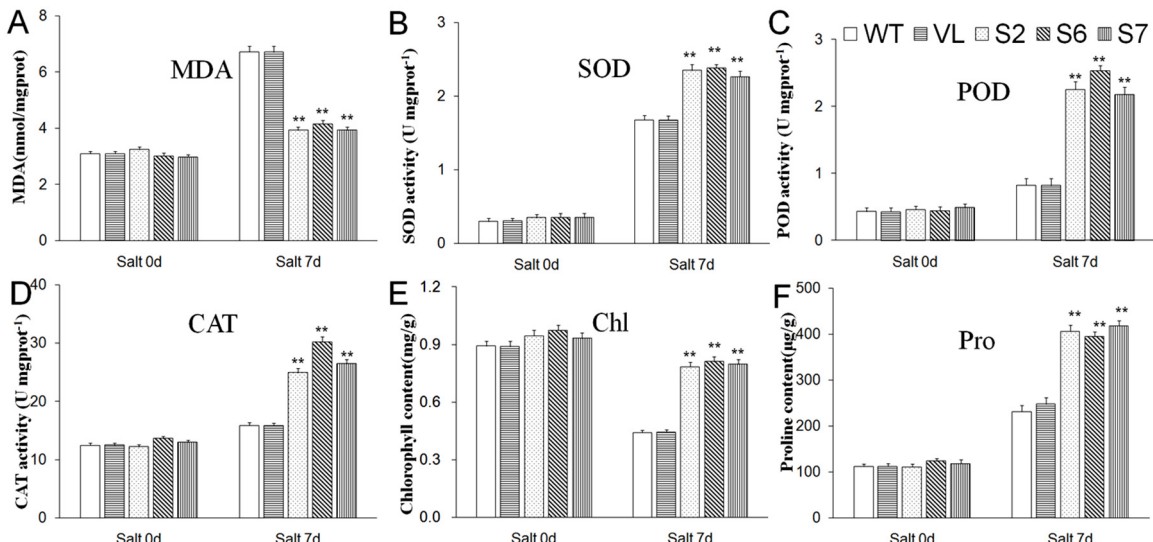

**Figure 8.** The levels of malondialdehyde content ((**A**), MDA); superoxide dismutase activity ((**B**), SOD); peroxidase activity ((**C**), POD); catalase activity ((**D**), CAT); chlorophyll content ((**E**), Chl) and proline content ((**F**), Pro) Proline in WT, VL (vacant-vector line) and *MbERF11*-OE (S2, S6, and S7) under control condition (Salt 0 d) and salt stresses (Salt 7 d). Asterisks above the error bars indicate an extremely significant difference between the transgenic and WT *Arabidopsis* with Student's *t* test (** $p \leq 0.01$). The level of each index in WT was as control.

## 4. Discussion

From the transcriptome analysis of *M. baccata* seedlings under cold and/or drought stresses, we found the *MbERF11* level was significantly up-regulated under both stresses. More importantly, through NCBI blast (https://blast.ncbi.nlm.nih.gov/Blast.cgi) of *MbERF11* gene, we found that the closest *Arabidopsis* ERF gene is *AtERF7*, which is a famous ERF TF gene involved in drought stress through ABA signal transduction [32]. Sequence homologous analysis showed that MbERF11 is a member of the ERF family (Figure 2A). All the ERF family includes one conserved ERF domain in the middle region [41]. These results showed that the ERF family genes are highly conserved during the evolution. ERF genes were widely distributed in apple, *A. thaliana*, pear, jujube, cucumber, tobacco, and rice, and were known to be involved in a variety of processes, including stress [10,12,18,21,32]. Subcellular localization has revealed that the MbERF11 is a nucleus localized protein (Figure 3), which is consistent with other ERF proteins [5,17,28,30,32]. Phylogenetic tree analysis shows that the relationship between MbERF11 and MdERF011 is the closest among 14 species. Among Arabidopsis ERF TFs, AtERF7 has the highest homology to MbERF11 (Figure 2B).

The expression level of *MbERF11* was more enriched in stem and root than in young leaf and mature leaf (Figure 4A). This expression pattern indicated that *MbERF11* may play an important role in organs that related to transport of stress signal. When dealt with cold, salt, and ethephon

treatments, the expression level of *MbERF11* in *M. baccata* was markedly affected. It is possible that *MbERF11* plays a key role in regulating stress response in *M. baccata*. Ethylene is considered as a signal of stress in plants [17,19,25,41], and ethylene treatment affected the expression of *MbERF11*. The changed expression level of *MbERF11* induced by cold and salt probably depends on the synthesis of endogenous ethylene.

When dealt with salt, cold, and ethephon treatments, the *MbERF11* expression level in root of *M. baccata* reached the highest level at 8 h, 12 h, and 2 h, respectively (Figure 4C), while in leaf, which got the maximum at 12 h, 24 h, and 4 h, respectively (Figure 4B). The results showed that the response rate to stresses, such as low temperature, salt stress and ethephon in root is faster than in leaf, indicating that the expression of *MbERF11* in root is more sensitive than that in leaf. This expression profile indicated that *MbERF11* may play a key role in the plants' stress signal transportation. Ethylene is a gaseous plant hormone that regulates many aspects of the plant life cycle, including seed germination, leaf senescence, fruit ripening, abscission, as well as biotic and abiotic stress responses [42–44]. Consequently, a higher concentration of ethylene in plants may be a signal to regulate plant stress response. This conclusion was consistent with the cold resistance mechanism of *VaERF057* [45]. Ethylene had been proposed to protect mitochondrial activity in *A. thaliana* under cold stress [46]. *GmERF7* had been confirmed to regulate the expression of stress-related genes through regulating the content of ethylene, thereby improving the salt tolerance [47]. Therefore, the increased expression level of *MbERF11* induced by cold and salt stresses may depend on the biosynthesis of ethylene.

The expression of *AtERF7* can be induced by ethylene, ABA and JA [32]. Abiotic stresses could induce the expression of *AtERF71*, *AtERF73*, *RAP2.1*, *RAP2.2*, and *RAP2.3* [48–50]. These results were consistent with our research (*MbERF11* expression level can be induced by ethylene, salt, and cold treatments). The expression level of *GmERF3* increased when dealt with abiotic stresses such as drought and high salinity. External factors such as ethylene and other hormone treatments could also change the expression level of *GmERF3*. However, low temperature stress had little effect on *GmERF3* expression [51]. The expression level of *MbERF11* gene in young leaf and new root was also significantly affected by cold, salt, and ethephon treatments (Figure 4B,C). Low temperature and salt stress could also increase the ethylene levels and trigger cold and salt stress responses in plants [30,46]. Based on the previous studies and theories, we reckon that ethephon treatments induce stress responses, such as the increased expression of *MbERF11* in the above parts.

Overexpression of *MbERF11* enhanced the tolerance to both cold and salt stresses in transgenic *A. thaliana*. The levels of chlorophyll, proline, MDA and antioxidant enzymes can be used to indicate the damage extent from stress [52,53]. The higher MDA content indicates higher degree of membranous peroxidation of the plant cells and the more serious damage to the cell membrane [2]. The proline content in WT *A. thaliana* increased after exposure to low temperature stress. Low temperature stress caused the destruction of chloroplast and the yellowing of plants. Hence, chlorophyll content is one of the important indicators of whether the plant is subjected to adverse stress [54]. The antioxidant enzyme system in plant plays an important role in resisting external environmental stress. They can inhibit the accumulation of free radicals, thereby reducing the occurrence of oxidative damage and lethal effects. The above effects allowed a variety of biochemical metabolic activities in cells to proceed normally. Overexpression of *MbERF11* enhanced the tolerance to cold and salt stresses in transgenic *A. thaliana* (Figures 5B and 7A), also led to increased activities of SOD, POD, and CAT, contents of proline and chlorophyll, decreased MDA content, especially when dealt with stresses (Figures 6 and 8). It is possible that *MbERF11* could increase cold and salt tolerance through changing these physiological indicators in transgenic *A. thaliana* under stress.

These results indicate that the *MbERF11* may be an upstream regulatory gene for stress resistance, and the overexpression of *MbERF11* gene may enhance the cold and salt tolerance of *A. thaliana.* More works need to be done to further verify the function of *MbERF11* through heterologous expression in *Arabidopsis* mutants (*AtERF7* gene deletion). Clarifying the role of the different domains of *MbERF11*

in stress response will be helpful in breeding stress-resistant *Malus* by gene transfer. Further experiments are required to identify other functions of *MbERF11* gene.

## 5. Conclusions

In the present study, a new ERF gene was isolated from M. baccata and named as MbERF11. Subcellular localization showed that MbERF11 protein was located to the nucleus. When MbERF11 was introduced into A. thaliana, it increased the levels of proline and chlorophyll, and improved the activities of SOD, POD, and CAT, but decreased MDA content, especially under cold and salt treatments. Taken together, our results suggest that MbERF11 plays an important role in response to cold and salt stress by enhancing the capability of scavenging ROS.

**Author Contributions:** D.H. and W.L. designed the experiments; D.H., J.H., G.Y., and S.W. performed the experiments; T.X. and J.H. analyzed the data; D.H. and W.L. wrote the manuscript. All authors have read and agreed to the published version of the manuscript.

**Funding:** This research was funded by by University Nursing Program for Young Scholars with Creative Talents in Heilongjiang Province (UNPYSCT-2017004), National Natural Science Foundation of China (31301757), Postdoctoral Scientific Research Development Fund of Heilongjiang Province, China (LBH-Q16020, LBH-Z15019), the Natural Science Fund Joint Guidance Project of Heilongjiang Province (LH2019C031) and the Open Project of Key Laboratory of Biology and Genetic Improvement of Horticultural Crops (Northeast Region), Ministry of Agriculture and Rural Affairs (neauhc201803).

**Conflicts of Interest:** The authors declare no conflicts of interest.

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
