# Peer review of "An ERF Transcription Factor Gene from Malus baccata (L.) Borkh, MbERF11, Affects Cold and Salt Stress Tolerance in Arabidopsis"

_forests, doi:10.3390/f11050514_

Round 1
Reviewer 1 Report
Han and colleagues, characterized in this work the MbERF11, a new ethylene response factor of Malus baccata. They perform a phylogenetic analysis of the gene, show that its expression responds to abiotic stress, that its protein seems to be nuclear-localized, and finally characterized Arabidopsis thaliana lines expressing MbERF11 suggesting a role for this gene in response to different abiotic stress.
Experiments are in general well designed and performed, but writing is poor. I recommend authors to hire the services of any professional translation company and revise deeply the whole writing of the paper.
Major concerns:
Besides the writing, I have two major experimental concerns about this work. The first relates to the motivation for studying MbERF11. Authors need to state clearly in the paper the reasons for choosing MbERF11 in their work. In my opinion, there is not a clear argumentation in the text about the initial motivation for studying this gene: is it because their closest homologs have been shown to be involved in stress tolerance? In which species? The second, is related with the previous one. Authors state in the article that ERF genes are widely distributed among plants, including Arabidopsis thaliana. But, I cannot find in the paper any reference to the Arabidopsis paralog or closet homolog to MbERF11, and that is in my opinion the mean weakness of this work. Do loss of function mutants for these paralogs exist in Arabidopsis or any other transformable plant model systems? In case they do, are they affected in abiotic stress responses? And if that is the case, could the heterologous expression of MbERF11 rescue their phenotypes? This would be a stronger proof of MbERF11 being involved in mediating abiotic stress response, that just overexpression in the Arabidopsis wildtype background.
Authors should address these concerns, and perform the experiments with the Arabidopsis mutants affected in paralogs, in case they exist, or at least rewriting some parts of the paper discussing the convenience of performing these experiments as future prospects.
Minor concerns:
Next, I suggest changes, clarifications and corrections that must be carried out to improve the paper (I indicate the sections´ paragraphs including issues to address):
Abstract
“Ethylene response factors, a class…”. Review writing, something is missing in this sentence.
Introduction
Paragraph 2:
“These were approximately 145 AP2/ERF transcription factors have been isolated in A. thaliana [11]” Check writing.
Paragraph 3:
“For DREB subfamily, the 14th and 19th amino acids are valine and glutamic acid, respectively. For ERF subfamily, which are alanine and aspartate.” Is this refered to the whole protein or to the AP2/ERF domain? Clarify, and check writing and grammar.
“The MdERF1/2 genes in M. domestica are associated with fruit ripening [30].” Since it is the first time that Malus domestica is quoted, use the complete name.
Materials and methods
Section 2.4. A. thaliana transformation. Please, specify in Table 1 the sequence of primers containing the SacI and BamHI cut sites used to generate the expression construct.
Section 2.5. Specify what VL stands for, since it is the first time that is quoted.
Results
Section 3.1. “The ProtParam analysis showed…” Add a reference or a webpage for ProtParam.
Section 3.2. Why is not included any Arabidopsis gene in the phylogenetic analysis? Please, include at least the closest homolog, if a clear paralog doesn’t exist.
Add a reference or webpage for DNAman.
Change “homologous tree” by “phylogenetic tree”.
Figure 2. Letter A should not be inside the figure. A and B panels are just touching each other, please separate. Indicate in figure 2 legend, which mean the shadow colors of aminoacids. Add a reference or webpage for MEGA 7.
Section 3.4
“…, and ethephon (500 L/L)” clarify what is ethephon adding “an ethylene production inducer”.
In my opinion, is better to use “young leaf”, instead of “new leaf”, to refer to a recently emerged leaf.
Figure 4. Some statistical differences look a little bit strange: young leaf, 2 h ethephon treatment P>0.01? Bars for cold 4 h and salt 8 h are taller compared to control and P>0.05. Moreover, there is no asterisk in root 12 h salt treatment. However, the different is quite important compare to the control, and error bars seem not to be quite large. Please, check, and additionally clarify in legend what the error bars represent (standard deviation?).
Section 3.5
It would have been better to use RT-qPCR, instead of just RT-qPCR, to demonstrate that the transgenic lines overexpress MbERF11 and that is the case, try to correlate the level of expression with the strength of the abiotic stress responses in different transgenic lines. Can authors perform this experiment?
Paragraph 2. What do authors mean by a “better phenotype”? I supposed that transgenic plants look healthier than control plants. Please, rewrite.
Paragraph 3. Define here, or in the Materials and methods section, how survival rate is determined.
Section 3.6
As in the previous section, define how survival rate is determined.
Figure 7B, it should be Salt stress instead of Cold stress.
Discussion
Paragraph 1. “ERF genes”, use italics when referred to genes.
Author Response
Han and colleagues, characterized in this work the MbERF11, a new ethylene response factor of Malus baccata. They perform a phylogenetic analysis of the gene, show that its expression responds to abiotic stress, that its protein seems to be nuclear-localized, and finally characterized Arabidopsis thaliana lines expressing MbERF11 suggesting a role for this gene in response to different abiotic stress.
Experiments are in general well designed and performed, but writing is poor. I recommend authors to hire the services of any professional translation company and revise deeply the whole writing of the paper.
Response: We appreciate your valuable suggestions and advices on our manuscript. I think they are very helpful and important, and revisions had been made in the revised manuscript accordingly.
Here I would like to response the comments and add some explanations as follows.
The language of manuscript have also been revised.
Major Concerns:
Besides the writing, I have two major experimental concerns about this work. The first relates to the motivation for studying MbERF11. Authors need to state clearly in the paper the reasons for choosing MbERF11 in their work. In my opinion, there is not a clear argumentation in the text about the initial motivation for studying this gene: is it because their closest homologs have been shown to be involved in stress tolerance? In which species?
Response: Yes, after the transcriptome analysis of cold and drought stress on Malus baccata (L.) Borkh, we found the expression level of MbERF11 is significantly up-regulated under both stresses. Through NCBI blast (https://blast.ncbi.nlm.nih.gov/Blast.cgi) of MbERF11, we found that the closest Arabidopsis ERF gene is AtERF7, a famous ERF transcription factor gene involved in drought stress through ABA signal transduction. So we dicided to research the function of MbERF11 in abiotic stresses.
The second, is related with the previous one. Authors state in the article that ERF genes are widely distributed among plants, including Arabidopsis thaliana. But, I cannot find in the paper any reference to the Arabidopsis paralog or closet homolog to MbERF11, and that is in my opinion the mean weakness of this work.
Response: Yes, we accepted your suggestion and added this part in the revised MS.
Do loss of function mutants for these paralogs exist in Arabidopsis or any other transformable plant model systems? In case they do, are they affected in abiotic stress responses? And if that is the case, could the heterologous expression of MbERF11 rescue their phenotypes? This would be a stronger proof of MbERF11 being involved in mediating abiotic stress response, that just overexpression in the Arabidopsis wildtype background.
Authors should address these concerns, and perform the experiments with the Arabidopsis mutants affected in paralogs, in case they exist, or at least rewriting some parts of the paper discussing the convenience of performing these experiments as future prospects.
Response: Yes, it is a very idea to research the MbERF11 function through heterologous expression in Arabidopsis mutants. We will do this work in the future. This part was added in the discussion of revised MS.
Minor concerns:
Next, I suggest changes, clarifications and corrections that must be carried out to improve the paper (I indicate the sections´ paragraphs including issues to address):
Abstract
“Ethylene response factors, a class…”. Review writing, something is missing in this sentence.
Response: Yes, we have revised this sentence.
Introduction
Paragraph 2:
“These were approximately 145 AP2/ERF transcription factors have been isolated in A. thaliana [11]” Check writing.
Response: Yes, we have revised this sentence.
Paragraph 3:
“For DREB subfamily, the 14th and 19th amino acids are valine and glutamic acid, respectively. For ERF subfamily, which are alanine and aspartate.” Is this refered to the whole protein or to the AP2/ERF domain? Clarify, and check writing and grammar.
Response: Yes, we accepted your suggestion and revised this sentence.
“The MdERF1/2 genes in M. domestica are associated with fruit ripening [30].” Since it is the first time that Malus domestica is quoted, use the complete name.
Response: Yes, we accepted your suggestion and used the complete name in the revised MS.
Materials and methods
Section 2.4. A. thaliana transformation. Please, specify in Table 1 the sequence of primers containing the SacI and BamHI cut sites used to generate the expression construct.
Response: Yes, the primers was added in the Table 1 of revised MS.
Section 2.5. Specify what VL stands for, since it is the first time that is quoted.
Response: Yes, the VL stands for Vacant-vector line, which in 2.5 of Materials and methods.
Results
Section 3.1. “The ProtParam analysis showed…” Add a reference or a webpage for ProtParam.
Response: Yes, the webpage for ProtParam was added in the revised MS.
Section 3.2. Why is not included any Arabidopsis gene in the phylogenetic analysis? Please, include at least the closest homolog, if a clear paralog doesn’t exist.
Response: For the Lower homology (even the most homologous protein AtERF7), Arabidopsis gene was not included in the phylogenetic analysis in the first manuscript.
But we accepted your suggestion and added the Arabidopsis ERF7 for phylogenetic analysis in the revised MS.
Add a reference or webpage for DNAman.
Response: Yes, a reference have been added here.
Change “homologous tree” by “phylogenetic tree”.
Response: Yes, “homologous tree” have been changed into “phylogenetic tree”.
Figure 2. Letter A should not be inside the figure. A and B panels are just touching each other, please separate. Indicate in figure 2 legend, which mean the shadow colors of aminoacids. Add a reference or webpage for MEGA 7.
Response: Yes, we have revised these parts in the Figure 2 of revised MS.
Section 3.4
“…, and ethephon (500 μL/L)” clarify what is ethephon adding “an ethylene production inducer”.
In my opinion, is better to use “young leaf”, instead of “new leaf”, to refer to a recently emerged leaf.
Response: Yes, we clarified ethephon was added into water, and used “young leaf” to refer to a recently emerged leaf in the revised MS.
Figure 4. Some statistical differences look a little bit strange: young leaf, 2 h ethephon treatment P>0.01? Bars for cold 4 h and salt 8 h are taller compared to control and P>0.05. Moreover, there is no asterisk in root 12 h salt treatment. However, the different is quite important compare to the control, and error bars seem not to be quite large. Please, check, and additionally clarify in legend what the error bars represent (standard deviation?).
Response: Yes, we revised these parts according your suggestion.
Section 3.5
It would have been better to use RT-qPCR, instead of just RT-PCR, to demonstrate that the transgenic lines overexpress MbERF11 and that is the case, try to correlate the level of expression with the strength of the abiotic stress responses in different transgenic lines. Can authors perform this experiment?
Response: Yes, we accept your suggestion and confirmed the transformed lines with RT-qPCR methord.
Thanks a lot for your suggestion to measure “the level of expression with the strength of the abiotic stress responses in different transgenic lines”. It is a very good idea for us, and your suggestion will help us in our future research on related research.
Paragraph 2. What do authors mean by a “better phenotype”? I supposed that transgenic plants look healthier than control plants. Please, rewrite.
Response: Yes, we accept your suggestion and revised this part.
Paragraph 3. Define here, or in the Materials and methods section, how survival rate is determined.
Response: Yes, we have added the method of measurement the survival rate in MM 2.5.
Section 3.6
As in the previous section, define how survival rate is determined.
Response: Yes, we have added the method of measurement the survival rate in MM 2.5.
Figure 7B, it should be Salt stress instead of Cold stress.
Response: Yes, it is our mistake. We have corrected it in the revised MS.
Discussion
Paragraph 1. “ERF genes”, use italics when referred to genes.
Response: Yes, we accept your suggestion and revised this part.
Reviewer 2 Report
Manuscript review
The manuscript “An ERF Transcription Factor Gene from Malus baccata (L.) Borkh, MbERF11, Affects Cold and Salt Stress Tolerance in Arabidopsis” by Han et al. describes the characterization of the transcription factor MbERF11 in response to cold and salt stress. Mb stands for Malus baccata, a crab apple species, but authors do not explain the importance of this species, and how their research relates to the domesticated apple (Malus domestica).
Major points
For the most part, the research appears sound, but it was difficult for me to gauge its significance, because no context was given:
- Have orthologues of MbERF11 already been characterized in other species?
- If so, to what extend?
- what was similar, what was different?
- What about MdERF11, the orthologue in the closely related domesticated apple?
- If so, to what extend?
These points should be mentioned in the introduction (“what is already known”) and discussed in detail in the discussion section (“how do results of this study relate to previous findings? How do the current results further our understanding?”).
Also, neither in results nor discussion was any context given on why and how MbERF11 was chosen. Of all possible ERF genes, why did the authors decide to focus on this particular gene???
The language needs some revision; I made many small suggestions throughout the text, but probably didn’t catch all.
For the most part, I could easily follow the flow of the story, further supported by appropriate figures, but the discussion will need major workto more clearly interpret the findings.
qRT-PCR
My main concern with the qRT-PCR is the use of just one reference gene, and no mentioning if this reference gene has been tested for uniform expression under the conditions tested here.
In Table I, the purpose of the target gene primers is stated as qPCR, while the purpose of the reference gene primers is stated as RT-PCR; the purpose for both primer pairs should be stated as qPCR.
Elsewhere, I suggest to use the term qRT-PCR (for quantitative reverse-transcriptase PCR).
Also, authors state the use of 2-ΔΔCT for quantification, but this only works for amplification efficiencies close to 100% (E=2).
If amplification efficiency was lower, were amplification efficiencies of target and reference gene taken into account, as in the Pfaffl method?
Additional points
Figure 1 has a small mistake (a wrong letter is boxed).
The figure legend should be expanded so the reader understands why certain letters are marked, without having to search for that information in the corresponding text.
Figure 4 is a time-course experiment and would be better presented as a line graph, rather than as the current bar graph. The focus should be on the change of expression for each condition over time, which would be much easier to follow in a line graph.
Figure 7B is incorrectly labeled as “cold stress”; it should be “salt stress”.
Discussion needs work
While the results were easy to follow, the discussion was not as straight forward. Taking out some redundancy and working on clarity would benefit the discussion, in addition to the above mentioning need to put the findings in context (What was already known? How do the results from this study compare to other findings?).
Minor points
In addition, I indicated many minor points directly in the text.

Author Response
The manuscript “An ERF Transcription Factor Gene from Malus baccata (L.) Borkh, MbERF11, Affects Cold and Salt Stress Tolerance in Arabidopsis” by Han et al. describes the characterization of the transcription factor MbERF11 in response to cold and salt stress. Mb stands for Malus baccata, a crab apple species, but authors do not explain the importance of this species, and how their research relates to the domesticated apple (Malus domestica).
Response: Yes, we accept your suggestion and added the importance of Malus baccata in the revised MS.
Apple's entire gene sequence map was sequenced with Malus domestica as materials. Based on this sequence map, it has become much easier for researchers to isolate genes from other Malus plants.
Major points
For the most part, the research appears sound, but it was difficult for me to gauge its significance, because no context was given:
- Have orthologues of MbERF11 already been characterized in other species?
If so, to what extend? what was similar, what was different?
- What about MdERF11, the orthologue in the closely related domesticated apple?
These points should be mentioned in the introduction (“what is already known”) and discussed in detail in the discussion section (“how do results of this study relate to previous findings? How do the current results further our understanding?”).
Response: Yes, we accepted your suggestion and added this part in the revised MS (More details had been presented in the results of AtERF7 gene research).
Also, neither in results nor discussion was any context given on why and how MbERF11 was chosen. Of all possible ERF genes, why did the authors decide to focus on this particular gene???
Response: Yes, after the transcriptome analysis of cold and drought stress on Malus baccata (L.) Borkh, we found the expression level of MbERF11 is significantly up-regulated under both stresses. Through NCBI blast (https://blast.ncbi.nlm.nih.gov/Blast.cgi) of MbERF11, we found that the closest Arabidopsis ERF gene is AtERF7, a famous ERF transcription factor gene involved in drought stress through ABA signal transduction. So we dicided to research the function of MbERF11 in abiotic stresses. This part was added in the discussion of MS.
The language needs some revision; I made many small suggestions throughout the text, but probably didn’t catch all.
Response: Yes, we accepted your suggestion and revised the language of manuscript.
For the most part, I could easily follow the flow of the story, further supported by appropriate figures, but the discussion will need major work to more clearly interpret the findings.
Response: Yes, we accepted your suggestion and revised the discussion part of manuscript.
qRT-PCR
My main concern with the qRT-PCR is the use of just one reference gene, and no mentioning if this reference gene has been tested for uniform expression under the conditions tested here.
Response: Yes, we accepted your suggestion and revised this part in the revised MS.
In Table I, the purpose of the target gene primers is stated as qPCR, while the purpose of the reference gene primers is stated as RT-PCR; the purpose for both primer pairs should be stated as qPCR.
Elsewhere, I suggest to use the term qRT-PCR (for quantitative reverse-transcriptase PCR).
Response: Yes, we accepted your suggestion and revised it.
Also, authors state the use of 2-ΔΔCT for quantification, but this only works for amplification efficiencies close to 100% (E=2). If amplification efficiency was lower, were amplification efficiencies of target and reference gene taken into account, as in the Pfaffl method?
Response: Yes, we accepted your suggestion and uesd Pfaffl method to reanalyse the qRT-PCR results.
Additional points
Figure 1 has a small mistake (a wrong letter is boxed).
Response: Yes, we had corrected this mistake.
The figure legend should be expanded so the reader understands why certain letters are marked, without having to search for that information in the corresponding text.
Response: Yes, we accepted your suggestion and expanded the figure legend.
Figure 4 is a time-course experiment and would be better presented as a line graph, rather than as the current bar graph. The focus should be on the change of expression for each condition over time, which would be much easier to follow in a line graph.
Response: Yes, you are right, the line graph would be much easier to follow the expression level with one treatment. The current bar graph is easier to show the difference between treatment and control, we think this is also important.
Figure 7B is incorrectly labeled as “cold stress”; it should be “salt stress”.
Response: Yes, it is our mistake. We have corrected it in the revised MS.
Discussion needs work
While the results were easy to follow, the discussion was not as straight forward. Taking out some redundancy and working on clarity would benefit the discussion, in addition to the above mentioning need to put the findings in context (What was already known? How do the results from this study compare to other findings?).
Response: Yes, we accepted your suggestion and revised this part.
Minor points
In addition, I indicated many minor points directly in the text.
Response: First of all, thank you for your huge and meticulous revise opinions on our manuscript. We appreciate your valuable suggestions and advices on our manuscript. I think they are very helpful and important.
Round 2
Reviewer 2 Report
Manuscript review
The revised manuscript addresses many, but not all of my concerns.
qRT-PCR
While the revisions sounded good when I first read the author’s responses; I am concerned that these were not fully implemented. Using two, rather than just one reference gene in the qRT-PCR and taking amplification efficiency into account did not change the results; how is that possible?
I understand that the authors do not want to repeat all qRT-PCR with additional reference genes; perhaps actin has been tested in various conditions and citing an appropriate paper could show that?
Why this ERF?
A major point that needed to be addressed is how the authors got interested in this particular ERF gene. This point is now addressed in the discussion, which is good, but should also be mentioned in the introduction.
Language
I made many small revisions in the text, but most of those were not implemented. So either hire a professional editor, or carefully improve the writing yourself. I again revised a little in the current manuscript, but not as much as I did previously.
The writing needs to be improved!

Author Response
The revised manuscript addresses many, but not all of my concerns.
Response: We appreciate your valuable suggestions and advices on our manuscript. I think they are very helpful and important, and revisions have been made in the revised manuscript accordingly.
Here I would like to response the comments and add some explanations as follows.
qRT-PCR
While the revisions sounded good when I first read the author’s responses; I am concerned that these were not fully implemented. Using two, rather than just one reference gene in the qRT-PCR and taking amplification efficiency into account did not change the results; how is that possible?
I understand that the authors do not want to repeat all qRT-PCR with additional reference genes; perhaps actin has been tested in various conditions and citing an appropriate paper could show that?
Response: Yes, we accept your suggestion and added a reference in the revised MS.
Why this ERF?
A major point that needed to be addressed is how the authors got interested in this particular ERF gene. This point is now addressed in the discussion, which is good, but should also be mentioned in the introduction.
Response: Yes, we accept your suggestion and added this part in the introduction of the revised MS.
Language
I made many small revisions in the text, but most of those were not implemented. So either hire a professional editor, or carefully improve the writing yourself. I again revised a little in the current manuscript, but not as much as I did previously.
The writing needs to be improved!
Response: First of all, thank you very much for your professional and meticulous modification suggestion.
The language of manuscript have been revised according your suggestion.
